# Pathotype Characterization of *Plasmodiophora brassicae*, the Cause of Clubroot in Central Europe and Sweden (2016–2020)

**DOI:** 10.3390/pathogens11121440

**Published:** 2022-11-29

**Authors:** Nazanin Zamani-Noor, Ann-Charlotte Wallenhammar, Joanna Kaczmarek, Usha Rani Patar, Miloslav Zouhar, Marie Manasova, Małgorzata Jędryczka

**Affiliations:** 1Julius Kühn-Institute (JKI), Institute for Plant Protection in Field Crops and Grassland, D-38104 Braunschweig, Germany; 2Rural Economy and Agricultural Society, HS Konsult AB, Gamla vägen 5G, SE-702 22 Örebro, Sweden; 3Pathogen Genetics and Plant Resistance Team, Institute of Plant Genetics, Polish Academy of Sciences, 60-479 Poznań, Poland; 4Department of Plant Protection, Faculty of Agrobiology, Food and Natural Resources, Czech University of Life Sciences, 165 00 Praha-Suchdol, Czech Republic

**Keywords:** clubroot, pathotype, physiological race, pathogenicity, oilseed rape, *Brassica napus*, disease incidence, disease severity index, virulence, differential set

## Abstract

Clubroot, caused by *Plasmodiophora brassicae*, is a crucial oilseed rape disease worldwide. Information on the virulence of *P. brassicae* populations is essential to apply disease control with proper clubroot-resistant cultivars. In 2016–2020, 84 isolates of *P. brassicae* were collected in the Czech Republic (CZ), Germany (DE), Poland (PL), and Sweden (SW). Pathotypes were designated using 17 Brassica hosts, including the European Clubroot Differentials (ECD), Somé set, and clubroot-resistant oilseed rape cv. Mendel. According to the ECD set, virulence analyses differentiated the isolates into 42 pathotypes. The most common pathotypes were 16/31/31 (in DE, PL, and SW) and 16/06/12 (in CZ, DE, and PL). Six pathotypes were found according to the Somé set, including 1–4 pathotypes per country. P1 was most prevalent in DE, PL, and SW, while P3 was abundant in CZ, DE, and PL. The current study provides clear evidence for a shift towards increased virulence in *P. brassicae* populations compared to previous studies. Several isolates overcame the resistance of cv. Mendel and of *Brassica rapa* genotypes ECD 01 to ECD 04. Considering all investigated samples, significant negative correlations were found between clubroot incidence and the frequency of oilseed rape in crop rotation, as for clubroot incidence and soil pH.

## 1. Introduction

Clubroot, caused by soilborne obligate pathogen *Plasmodiophora brassicae* Woronin, is a constraint for oilseed rape (*Brassica napus* L.) production worldwide in Europe, Latin America, North America, Canada, Australia, Japan, and China [1,2,3,4,5,6,7,8,9,10,11,12,13,14]. The clubroot disease can severely damage oilseed rape and cause significant yield losses [13,15]. Therefore, developing and growing resistant oilseed rape cultivars is the most persuasive, economical, easy-to-use, and environmentally friendly control strategy for disease control. However, the *P. brassicae* pathogen is remarkably variable in virulence because it maintains evolving new pathotypes and changes in the distribution and frequency of previously existing pathotypes within individual fields. Consequently, these changes in the *P. brassicae* pathotype can overcome the resistance of previously clubroot-resistant cultivars, which results in unexpected disease outbreaks and further yield losses [10,16,17].

For decades, *P. brassicae*-pathotypes worldwide have been classified by evaluating their ability to infect single gene lines or host genotypes specified as a “differential set”. Several differential sets or classification systems have been established and used around the world [18,19,20], which can ease virulence comparisons between different regions or countries. For example, more recently, Řičařová et al. [9] conducted pathotype monitoring in the Czech Republic and Poland. They identified five pathotypes based on the system of Somé et al., seven pathotypes with the differentials of Williams, and 18 pathotypes according to the European Clubroot Differential (ECD) set [18,19,20]. Lüders et al. [21] and Zamani-Noor [17] also showed variation in pathotype distribution in different areas in Germany. In their studies, among several field isolates of *P. brassicae*, the majority, according to the classification system of Somé et al. [20], were pathotypes P1 and P3, and the minority were pathotypes P2, P4, and P5. Moreover, according to Buczacki et al. [19], detailed classification showed the dominance of 16/31/31, 16/14/30, and 16/14/31 populations among 20 distinct virulence patterns [17]. Additionally, several isolates were identified as moderately or highly virulent on the clubroot-resistant oilseed rape cultivars [17,21]. These new isolates were informally nominated as P1 (+), P2 (+), or P3 (+) because they are classified as P1, P2, or P3 on the differential system of Somé et al. [20] but (unlike the initial P1, P2 or P3) are highly virulent on the commercial clubroot-resistant oilseed rape cultivar Mendel. Further studies on the characterization of *P. brassicae* populations according to the ECD set and classification by the differential hosts of Somé et al. or Williams were conducted in central Europe [22]. The results revealed that the pathotype distribution varied between the European countries. In France, six different pathotypes were classified according to the set of Somé et al. [20]. According to the ECD set, 11 virulence groups were found in the UK. According to the differential set elaborated by Williams (1966), the most common pathotypes in central Europe were 4, 6, and 7. According to the system developed by Somé et al. (1996), P1 and P3 were the most frequent. Based on studies using the differential set elaborated by Buczacki et al. [19], the most frequent pathotypes in European *P. brassicae*-populations were ECD 16/31/31 and 16/14/31. While most of the *B. rapa* genotypes from the ECD set were resistant to the Czech, German, and Polish *P. brassicae* populations, several isolates found in the UK and Sweden, which could overcome the resistance of ECD 01, ECD 02, and ECD 03. In addition, some isolates were identified in Sweden that could overcome the resistance of all ECD *B. rapa*-hosts [22]. In Canada, Strelkov et al. [23] also reported new isolates of *P. brassicae,* which could overcome the resistance in clubroot-resistant canola in central Alberta, Canada. These isolates were classified as pathotype 5 on the differentials of Williams [18] but differed from previously characterized pathotype 5 strains based on their increased virulence on clubroot-resistant hosts. Furthermore, new virulent phenotypes were reported from clubroot-resistant canola fields in Alberta, leading to the development of a new pathotype classification system, the Canadian Clubroot Differential (CCD) set, to accommodate these new isolates [16].

In summary, frequent monitoring of the virulence spectra of *P. brassicae* isolates and identifying new pathotypes are crucial for effective disease management strategies through resistance breeding. In addition, the identification of virulent pathotypes is essential for the characterization of non-pathotype-specific resistance. Information on virulence is also appropriate in screening genotypes or breeding lines for determining resistance levels and using new resistance genes. Furthermore, monitoring the virulence of the clubroot pathogen population in different regions promotes understanding the genetic variability of host–pathogen interactions.

The main objectives of the current study were to (1) evaluate the distribution and prevalence of *Plasmodiophora brassicae* pathotypes in oilseed rape crops in the Czech Republic, Germany, Poland, and Sweden over five years (from 2016 to 2020); (2) identify the virulence of the collected isolates on the clubroot-resistant oilseed rape cv. Mendel, and (3) assess the possible correlations between the disease incidence in infested fields with crop rotation and soil pH.

## 2. Materials and Methods

### 2.1. Clubroot Survey and Collection of P. brassicae Isolates

Clubroot-infected root samples were collected from commercial oilseed rape fields by the authors and collaborators (farmers, oilseed rape breeders, and agricultural consultants) throughout Moravia in the Czech Republic, Germany, and Poland during the oilseed rape-growing seasons from 2016 to 2020. The collection and maintenance of isolates from the Czech Republic and Poland were conducted according to Řičařová et al. [9]; the isolates from Germany were obtained according to Zamani-Noor [17]. All Swedish isolates except one were collected from experimental field sites where clubroot-resistant cultivars were evaluated for disease severity and yield performance. The field sites were selected for the prevalence of *P. brassicae* DNA, according to Wallenhammar et al. [24]. The experimental setup was a randomized design with four replicates. Twenty-five plants were assessed for disease in each plot, with 100 plants for each treatment. In addition, one sample was collected from a farm field of a susceptible cultivar. The root samples were washed and stored frozen at −20 °C until they were sent to the laboratory at Julius Kühn-Institute (JKI), Braunschweig, Germany. In total, 84 isolates were studied: 10 isolates that originated from the Czech Republic (Figure 1A), 41 isolates from Germany (Figure 1B), 25 isolates from Poland (Figure 1C), and 8 isolates from Sweden (Figure 1D). 

The soil pH was measured by professional soil pH meters, Hanna Instruments HI 98128 in the Czech Republic, and a Draminski Ltd. soil pH-meter in Poland, respectively. In Germany, soil pH was measured by mixing air-dried soil with double-distilled water (1:1), shaking for one hour, and measuring the value with a pH meter (Seven Easy pH, Mettler-Toledo AG, Greifensee, Switzerland). In Sweden, soil pH was determined potentiometrically in water (1.0:2.5, weight/weight) by Euro Eurofins Agro Testing Sweden AB., Kristianstad, Sweden.

The information about each studied field concerned the geographical location, soil pH, rotation regime (number of years since the previous oilseed rape cultivation), and clubroot incidence, expressed as the percentage of plants infected in the field, mostly out of 100 randomly selected plants.

### 2.2. Virulence Phenotyping of Differential Sets

*Plasmodiophora brassicae* resting spores were extracted from the clubbed roots of each field, according to Řičařová et al. [9] and Zamani-Noor [17]. The resting spores originating from one field were regarded as one pathogen population. The pathotype classification of the *P. brassicae* populations was conducted on a set of 16 Brassica hosts that included the European Clubroot Differential (ECD) set [19] and Somé et al. [20]. Furthermore, each population was inoculated separately onto the clubroot-resistant oilseed rape cv. Mendel to evaluate the degree of virulence of the collected populations. Seeds of the ECD set were provided by the Genetic Resources Unit, Warwick Crop Centre, the University of Warwick (Wellesbourne, UK). Seeds of cultivar Mendel and the spring-oilseed rape cultivar Brutor (belonging to the Somé differential set [20]) were kindly provided by Norddeutsche Pflanzenzucht Hans-Georg Lembke KG and the Leibniz Institute of Plant Genetics and Crop Plant Research Genebank (IPK), respectively.

Inoculation and growth conditions for the plants after inoculation were the same as those described in Řičařová et al. [9] and Zamani-Noor [17]. Infection types were scored 35 days after inoculation using a 0–3 scale: 0—no galling, 1—a few small galls, 2—moderate galling on the main and lateral roots, and 3—severe galling and the root is deformed. The disease severity index (DSI) was calculated from each infection type according to Horiuchi and Hori [25], which was modified by Strelkov et al. [26]. DSI data were converted to susceptible or resistant to describe the resistant-susceptible patterns for each plant genotype. A differential line was considered resistant when DSI was lower or equal to 25% or susceptible when DSI was higher than 25% [20].

### 2.3. Pathotype Identification, Frequency, and Distribution

The pathotype of each population was identified based on its virulence–avirulence patterns on the differential sets. The pathotypes overcoming the resistance of cv. Mendel were designated with an additional (+), as suggested by Zamani-Noor et al. [27]. The frequency of a pathotype in each country was calculated as the percentage of pathotypes with a specific virulence. Finally, the distributions of detected pathotypes were summarized by regions and the whole country.

### 2.4. Statistical Analyses

Comparisons between different genotypes and disease severity indexes were accomplished by analysis of variance (ANOVA) using Fisher’s least significant difference (LSD) and considered significant at *p* ≤ 0.05 in Statistica version 9.1 (Stat Soft, Inc., Tulsa, OK, USA). In addition, Spearman’s rank and Pearson’s correlation coefficients were calculated to analyze the relationship between soil pH and the rotation regime with clubroot disease incidence (%) in naturally *P. brassicae*-infested fields. Graphs describe box–whisker plots with median, 25%, 75% quartiles, and minimum and maximum values. Outliers and extremes are not shown in box–whisker graphs.

## 3. Results

Information on the geographical location, soil pH, rotation regime, and clubroot incidence (percentage of plants infected in the field) is provided in relevant tables, including data for the Czech Republic (Table 1), Germany (Table 2), Poland (Table 3), and Sweden (Table 4).

### 3.1. Clubroot in the Czech Republic

#### 3.1.1. Disease Occurrence and Severity

Fields in the Czech Republic have been regularly monitored for the presence of clubroot disease with more emphasis since the last decade. Clubroot disease incidence ranged between 3 to 27%. It was observed that 60% of the fields had disease incidence below 10% while the others had a moderate disease incidence in an average range between 20 and 27%.

**Table 1 pathogens-11-01440-t001:** Origin of *Plasmodiophora brassicae* field isolates, the related field information, and the results of pathotype classification. Isolates were collected from 10 clubroot-infested fields in Moravia, the Czech Republic, from 2016 to 2020.

					Pathotype Classification ^3^		
Isolate	Region ^1^	Soil pH	Frequency of OSR in a Rotation (Years)	Mean Value of DI per Region in Clubroot-Infested Fields (%) ^2^	ECD Code	Somé 1996	DSI on OSR cv. Mendel
1	MR	5.61	4	11.91 ± 9.23	16/10/00	P4	0
2	MR	6.80	4		16/15/29	P2	25
3	MR	6.20	4		16/15/08	P6	20
4	MR	5.82	3		16/06/25	P3	12
5	MR	5.65	4		16/06/12	P3	0
6	MR	5.73	4		16/14/24	P3	22
7	MR	6.77	4		16/10/00	P3	4
8	MR	6.74	4		16/06/12	P3	12
9	MR	6.21	3		16/04/00	P4	0
10	MR	6.76	4		16/14/13	P3	9

^1^ Region in the Czech Republic: MR: Moravia ^2^ Mean value of the disease incidence (DI) in clubroot-infested fields in the region. The incidence of disease in each clubroot-infested field was estimated by randomly choosing 100 plants (10 plants at 10 sites in the field). The presence of any swelling or gall formation on the root was assumed to be proof of *P. brassicae* infection. Several galls on roots were cut to check for the presence/absence of gall weevil *Ceutorhynchus assimilis*. ^3^ Classification was done according to the differential systems of the European Clubroot Differential set (ECD; [19]) and Somé et al. [20]. A cut-off point of the disease severity index (DSI) of 25% was used to classify plant reactions as resistant or susceptible [20].

In eight out of ten fields, oilseed rape was grown with a crop rotation interval of four years, while in two fields; the crop rotation interval was three years (Figure 2A). There was a significantly strong but negative correlation (−0.696, *p* = 0.025) between the frequency of oilseed rape in a crop rotation and the average disease incidence, indicating a significant relationship. The soil pH in the fields was mainly recorded as moderately acidic ranging from 5.61 to 6.80. Soil pH was lower than 6.5 in six fields, whereas in four fields, soil pH ranged from 6.5 to 6.8 (Table 1). There was a moderately positive but non-significant correlation (r_s_ = 0.2336; *p* = 0.5159) between soil pH and average disease incidence (Figure 2B). Moreover, a similarity in the average disease incidence was found between the two fields with a soil pH of 5.82 (DSI = 21.6%) and 6.8 (DSI = 21.5%), despite a large difference in the soil pH value. 

#### 3.1.2. Pathotypes of *Plasmodiophora brassicae* in the Czech Republic

According to the pathotype designation system of Somé et al. [20], pathotype P3 was prevalent (60% of the isolates), followed by pathotype P4 (20%). The other observed pathotypes were P2 and P6 in one isolate each (10%) (Table 1). 

Following the ECD system [19], nine unique pathotypes were reported in the Czech isolates. The pathotypes 16/06/12 and 16/10/00 were observed in two isolates out of ten, and the other pathotypes were found in one isolate each (Table 1). ECD 05 (cv. Granaat) was susceptible to all isolates, whereas none of the isolates could overcome the resistance of the ECD 01 to ECD 04 accessions of *B. rapa*. Six reactions were recorded in response to the *B. napus* accessions (ECD 06 to ECD 10) with scores 4, 6, 10, 14, and 15 (Table 1). Most isolates (80%) showed strong virulence on ECD 07 and ECD 08. On the other hand, none of the isolates could overcome the resistance of ECD 10 (cv. Wilhelmsburger). The most virulent isolates could overcome the resistance of all of the *B. napus* genotypes (ECD 06 to ECD 09) except ECD 10. Five out of ten isolates studied overcame the resistance of ECD 09 (cv. New Zealand). In the case of the *B. oleracea* accessions, diversity was recorded in the responses of the isolates. Seven reactions were recorded with scores of 0, 8, 12, 13, 24, 25, and 29. All of the genotypes of *B. oleracea* remained resistant to isolates 1, 7 and 9 (Table 1). ECD 12 (cv. Bindsachsener) was resistant to all but one isolate. On the other hand, six isolates were virulent in the cultivar ECD 14 (cv. Septa), while four and three of the isolates showed virulent reactions on ECD 13 and ECD 15. The most virulent isolate could infect four out of five genotypes of *B. oleracea* of the ECD set. All isolates were avirulent (DSI < 25) to the clubroot-resistant cultivar ‘Mendel’; however, only three isolates showed no symptoms (DSI = 0), while isolates 3 and 4 showed DSI ranging from 20 to 25% and from 4 to 12%, respectively.

### 3.2. Clubroot in Germany

#### 3.2.1. Disease Occurrence and Severity

During the five-year clubroot study in Germany, 41 new clubroot-infested fields were identified in seven federal states. Clubroot disease incidence varied within these fields from 5% to 80%. Fifteen fields showed a low disease incidence (DI ≤ 30%), 19 fields showed moderate clubroot incidences (30% < DI < 60%), and seven fields exhibited a high disease incidence (DI ≥ 60%). A mean value of DI (%) for each federal state is given in Table 1. The information on field data showed that oilseed rape was grown every 2^nd^ year in 9% of the fields and every 3rd year in 51% of the fields, while the interval exceeded three years in 40% of the fields (Table 2).

**Table 2 pathogens-11-01440-t002:** Origin of *Plasmodiophora brassicae* field isolates, the related field information, and the results of pathotype classification. Isolates were collected from 41 clubroot-infested fields in seven federal states in Germany from 2016 to 2020.

					Pathotype Classification ^3^		
Isolate	Federal State ^1^	Soil pH	Frequency of OSR in a Rotation (Years)	Mean Value of DI per Region in Clubroot-Infested Field (%) ^2^	ECD Code	Somé 1996	DSI on OSR cv. Mendel
1	HE	6.39	5	36.9 ± 15.1	17/31/31	P1	9
2	HE	5.89	4		19/31/31	P1 (+)	100
3	HE	6.44	3		16/31/31	P1 (+)	100
4	HE	5.48	2		24/31/27	P1 (+)	95
5	HE	5.73	4		16/31/31	P1 (+)	100
6	HE	6.96	4		16/28/30	P5	0
7	HE	6.19	3		16/03/31	P2	12
8	HE	5.11	3		17/22/31	P5	17
9	NI	5.20	2	40.5 ± 18.7	16/23/30	P1 (+)	65
10	NI	5.72	4		16/31/31	P1 (+)	75
11	NI	6.30	3		16/14/21	P3	0
12	NI	6.02	4		16/012/31	P3	5
13	NI	6.39	2		16/31/31	P1	10
14	NI	6.23	3		17/31/31	P1 (+)	90
15	NI	5.76	5		16/31/31	P1	10
16	NI	5.78	5		16/31/31	P1	0
17	NI	6.90	5		16/30/31	P3	22
18	NI	6.20	3		17/31/31	P1 (+)	60
19	NI	6.60	3		16/06/12	P3	12
20	MV	6.73	2	51.1 ± 24.7	16/31/12	P1	87
21	MV	6.25	3		16/14/30	P3	0
22	MV	5.90	3		16/31/30	P1	25
23	MV	6.30	5		16/31/12	P1	5
24	MV	5.80	3		16/31/30	P1	15
25	MV	6.70	3		16/31/31	P1	10
26	MV	6.20	3		16/31/30	P1	15
27	MV	5.86	3		16/31/31	P1	8
28	MV	6.40	4		16/31/30	P1	12
29	NW	5.51	4	22.5 ± 3.5	17/31/31	P1	8
30	NW	6.45	4		16/15/31	P3 (+)	80
31	RP	6.50	5	10.0 ± 0.0	16/06/30	P3	0
32	SN	5.71	4	55.0 ± 14.1	16/02/30	P3	15
33	SN	6.30	3		16/31/31	P1	25
34	SH	5.27	3	41.3 ± 13.6	16/31/31	P1 (+)	100
35	SH	6.80	4		16/31/30	P1	18
36	SH	6.50	3		16/31/30	P1	12
37	SH	6.15	3		16/14/31	P3	0
38	SH	6.25	3		16/31/30	P1	35
39	SH	6.36	3		16/31/31	P1	25
40	SH	5.72	3		16/31/30	P1 (+)	100
41	SH	6.75	3		16/02/30	P3 (+)	45

^1^ Federal States in Germany: HE: Hesse; NI: Lower Saxony; MV: Mecklenburg-Vorpommern; NW: North Rhine-Westphalia; RP: Rhineland-Palatinate; SN: Saxony; SH: Schleswig-Holstein. ^2^ Mean value of disease incidence (DI) in clubroot-infested fields in each federal state. The incidence of disease in each clubroot-infested field was estimated by randomly choosing 50 plants. The presence of any swelling or gall formation on the root was assumed as proof of *P. brassicae* infection. ^3^ Classification was done according to the differential systems of the European Clubroot Differential set (ECD; [19]) and Somé et al. [20]. A cut-off point of the disease severity index (DSI) of 25% was used to classify plant reactions as resistant or susceptible [20].

The frequency of oilseed rape cultivation in the rotation strongly influenced the disease incidence in the field. The highest incidences (DI ≥ 60%) were observed in the fields with more frequent cropping of oilseed rape in the rotation (Figure 3A).

Moreover, the field information revealed that, on average, lower soil pH coincided with higher disease incidence. Across the soil samples from the clubroot-infested fields, pH values ranged from 5.11 to 6.97. Soil pH frequency distribution indicated that 73% of the soil samples had a pH of 5.50–6.50 (Table 2). The frequency decreased to 36.5% and 17% when the soil pH values ranged from 5.1 to 6.0 or 6.6 to 7.0, respectively (Table 2). A weak but insignificant negative correlation was found between soil pH and the disease incidence of infested fields (Figure 3B).

#### 3.2.2. Pathotypes of *Plasmodiophora brassicae* in Germany

The pathotype classification was conducted under controlled greenhouse conditions twice with comparable results (correlation data not shown). Significant differences were also noticed in the virulence of the 41 *P. brassicae* field isolates and the variation in pathotype distribution in different areas. The detailed result of pathotype classification according to the ECD set of Buczacki et al. [19] and Somé et al. [20] is given in Table 2.

Typical clubroot galls were well developed on susceptible plants such as *B. rapa* var. *pekinensis* (ECD 05) under greenhouse conditions at 35 dpi. The resistant *B. rapa* differential genotypes, ECD 01 to ECD 04, were not infected by 83% of the *P. brassicae* isolates (Table 2). The exceptions were for ECD 01, susceptible to isolates 1, 2, 8, 14, 18, and 29, and ECD 02 and ECD 04, susceptible to isolates 2 and 4, respectively (Table 2). In contrast, most of the *P. brassicae* isolates were virulent *on B. napus* cultivars, ECD 06–ECD 10, and *B. oleracea* differentials, ECD 11–ECD 15 (Table 2*). Brassica napus* cv. ‘Brutor’ was heavily infected by all *P. brassicae* isolates. Among all collected isolates, 19 virulence patterns were found according to the ECD differential system of Buczacki et al. (1975). The predominant isolates were 16/31/31, 16/31/30, and 17/31/31, accounting for 27%, 20%, and 10% of all isolates. Four pathotypes were identified based on the isolate’s virulence to the set of Somé et al. [20]. The majority of pathotypes were P1 and P3, while P2 and P5 were in the minority (Table 2). Moreover, significant differences were observed in the virulence of different *P. brassicae* isolates on clubroot-resistant oilseed rape cv. Mendel (Table 2). Fourteen isolates were identified as virulent (DSI > 25), with disease severity indices ranging from 35% to 100% (Table 2).

### 3.3. Clubroot in Poland

#### 3.3.1. Disease Occurrence and Severity

In Poland, the fields with oilseed rape plants bearing clubroot symptoms were constantly observed each year from 2016 to 2020. Disease symptoms were present in most of the regions of intensive oilseed rape cultivation. The disease incidence varied greatly between regions, ranging from 7.0% to 60.3% (Table 3). The disease was mainly patchy, with some spots in the field without symptoms on plants and the other sites showing all plants with galls on roots (data not shown). On average, the DI was below 30 (80% of the fields). Disease incidence exceeding 60% was noted in 12% of the fields, and 30% < DI < 60% was recorded in 8% of the fields (Table 3).

Cutting the galls with a sharp knife showed no insects or insect wounds inside the galls, so the visual inspection attributed the disease symptoms solely to clubroot. These observations were fully confirmed in the laboratory. Koch’s postulates were confirmed by the isolation of the resting spores in collected galls, which caused infection of the susceptible accession of the Buczacki et al. [19] pathotyping system (*B. rapa* cv. Granaat; ECD 05).

**Table 3 pathogens-11-01440-t003:** Origin of *Plasmodiophora brassicae* field isolates, the related field information, and the results of pathotype classification. Isolates were collected from 25 clubroot-infested fields in 12 regions in Poland from 2016 to 2020.

			Frequency of OSR in a Rotation (Years)	Mean Value of DI per Region in Clubroot-Infested Field (%) ^2^	Pathotype Classification ^3^		
Isolate	Region ^1^	Soil pH			ECD Code	Somé 1996	DSI on OSR cv. Mendel
1	KP	6.6	4	25	21/31/15	P1	22
2	LP	6.7	3	25	31/15/31	P2 (+)	81
3	LS	6.5	2	11.5 ± 4.9	16/14/15	P3	12
4	LS	6.6	3		16.06.2012	P3	4
5	LU	5.9	3	26.5 ± 6.4	16/31/31	P1	20
6	LU	6.7	3		31/15/31	P2 (+)	100
7	LU	6.6	3		31/31/15	P1 (+)	75
8	MZ	6.5	4	7	16.06.2029	P4	17
9	OP	6.7	3	29 ± 19.8	31/31/15	P1 (+)	36
10	OP	6.5	4		31/15/15	P2	0
11	PD	6.5	4	16	21/31/31	P1 (+)	86
12	PM	6.8	3	22.5 ± 13.4	16/15/31	P2	22
13	PM	6.7	2		16/14/15	P3	31
14	SC	6.2	3	20.7 ± 8.0	31/31/31	P1 (+)	82
15	SC	6.5	3		28/15/09	P2 (+)	78
16	SC	6.8	4		31/31/31	P1 (+)	100
17	SL	6.1	3	14	20/15/15	P2	0
18	VM	5.9	2	27.6 ± 17.3	16/15/08	P2	0
19	VM	6.4	3		16/15/31	P2 (+)	100
20	VM	5.9	4		26/15/31	P2 (+)	100
21	VM	6.6	3		31/15/31	P2 (+)	93
22	VM	6.2	3		31/31/31	P1 (+)	75
23	WP	6.3	1	60.3 ± 8.08	30/31/24	P1 (+)	43
24	WP	6.1	3		30/31/26	P1 (+)	100
25	WP	6.3	3		31/31/31	P1 (+)	100

^1^ Regions in Poland: KP—Kujavia-Pomerania, LP—Lower Poland, LS—Lower Silesia, LU—Lublin region, MZ—Mazovia, OP—Opole region, PD—Podlasie, PM—Pomerania, SC—Subcarpathia, SL—Silesia, WP—West Pomerania, VM—Varmia-Masuria; ^2^ Mean value of disease incidence (DI) in clubroot-infested fields in each region. The incidence of disease in each clubroot-infested field was estimated by randomly choosing 100 plants (20 plants at 5 sites in the field). The presence of swelling or gall formation on the root was assumed as proof of *P. brassicae* infection. Several galls on roots were cut to check for the presence/absence of gall weevil *Ceutorhynchus assimilis*. ^3^ Classification was done according to the differential systems of the European Clubroot Differential set (ECD; [19]) and Somé et al. [20]. A cut-off point of the disease severity index (DSI) of 25% was used to classify plant reactions as resistant or susceptible [20].

In the studied fields, oilseed rape was mainly grown every three years (56% of the fields), followed by a 4-year rotation (28%), and in 12% (3 fields), oilseed rape was grown every two years. In one field, oilseed rape was also grown in the preceding year, which coincided with the highest clubroot infestation among studied fields (69% on average). Pearson’s correlation coefficient between the frequency of oilseed rape in a rotation and the average disease incidence was 0.544, showing the considerable link between these parameters. A crop rotation of four years of the time interval of oilseed rape cultivation coincided with an increase in the incidence of 18% on average. In contrast, 2-year intervals resulted in disease levels that were nearly twice as high (Figure 4A).

Most of the soils had a pH ranging from 6.0 to 6.5 (48%) or 6.6 to 6.8 (40%), and only 12% (three fields) had a pH below 6.0 (Table 3). Pearson’s correlation between soil pH and disease incidence was much lower and negative (r_s_ = −0.183) and indicated that disease symptoms were observed in soils with both high and low pH (Figure 4B). There was a slight trend to find more plants with clubs in soils with a pH above 6.5 compared to soil with a pH below this value (Figure 4B). The time interval between the cultivation of oilseed rape in crop rotation was more strongly linked to clubroot incidence than soil pH (Figure 4A,B).

#### 3.3.2. Pathotypes of *Plasmodiophora brassicae* in Poland

According to the pathotype designation elaborated by Somé et al. [20], the dominating pathotypes in Poland (60%) could overcome the resistance of cv. Mendel, with 36% of the isolates belonging to P1 (+) and 24% isolates of the P2 (+) pathotype. The remaining 40% of the isolates belonged to the pathotypes P1–P4, ranging from 1–4 isolates (Table 3).

According to Buczacki et al. [19], pathotype designation resulted in 16 pathotypes, among which the most common were 31/31/31 and 31/15/31, with four and three isolates, respectively. The other pathotypes were represented by one or two isolates only (Table 3). Regarding the susceptibility of the *B. rapa* accessions present in the ECD differential set, there were seven types of responses, referred to as 16, 20, 21, 26, 28, 30, and 31 scores. This means that at least one and up to all (5) accessions of *B. rapa* were infected with the isolates studied. Out of the 25 isolates of *P. brassicae* obtained from Poland, 40% (10 isolates) caused symptoms on all five accessions, and 32% (8 isolates) were able to infect *B. rapa* cv. Granaat only, whereas the remaining 28% (7 isolates) could overcome the resistance of at least one or more of the remaining accessions of *B. rapa* in the ECD set (Table 3). Among the responses to *B. oleracea*, eight different reactions were observed, referred to as 8, 9, 12, 15, 24, 26, 29, and 31 scores. Again, the prevalent number of isolates (12) could cause symptoms on all five accessions (score 31), which constituted 48% of the isolates of *P. brassicae* from Poland. The second most common response (7 isolates, 28%) scored 15, which means that the only resistant accession was ECD15 (cv. Verheul). The lowest diversity was observed among the susceptibility of *B. napus* accessions, with 6, 14, 15, and 31 scores, two of which (31 and 15) represented most of the isolates, 11 (44%) and 10 (40%), respectively. The other responses of *B. oleracea* to the inoculation were caused by four isolates only (16%), and this means that the majority (4) or all (5) accessions of *B. oleracea* represented in the ECD differential set were susceptible to the *P. brassicae* isolates originating from Poland.

### 3.4. Clubroot in Sweden

#### 3.4.1. Disease Occurrence and Severity

Most Swedish isolates were collected from cultivar trials at experimental field sites where the prevalence of *P. brassicae* DNA was determined before seeding the experiment. The incidence of clubroot disease for susceptible cultivar mixtures ranged from 45% to 100%; the mean value of DI (%) for each region is given in Table 4. Isolates 5–8 were sampled in April; thus, DI for isolate 8 is underestimated as infected plants have likely disappeared during winter. The clubroot-resistant cultivars RNX 193207 and DK Plastic showed DI values of 42.5 and 55, indicating the deployment of resistance, whereas resistant cultivars Croozer and Crocodile showed a DI of 12.5 and 10, respectively. 

**Table 4 pathogens-11-01440-t004:** Origin of *Plasmodiophora brassicae* field isolates, the related field information, and the results of pathotype classification. Isolates were collected from four field experiments and from one clubroot-infested field in southern and central Sweden from 2017 to 2020.

			Frequency of OSR in a Rotation (Years)	Mean Value of DI per Region in Clubroot-Infested Fields (%) ^3^	Pathotype Classification ^4^		
Isolate ^1^	Region ^2^	Soil pH			ECD Code	Somé 1996	DSI on OSR cv. Mendel
1	NA	6.1	6	100 ± 0.0	16/23/28	P1	8
2	NA	6.4	6		16/23/29	P1	27
3	SC	6.5	4	47.56 ± 19.76	16/31/31	P1 (+)	47
4	SC	6.5	4		16/23/29	P1	22
5	SC	6.9	4		16/31/31	P1 (+)	75
6	SC	6.9	4		16/31/31	P1 (+)	57
7	SC	6.9	4		16/31/31	P1 (+)	63
8	SC	6.9	4		16/31/28	P1 (+)	43

^1^ Isolate 1: Susceptible cultivar mix (cvs Avatar, Dariot and Explicit); Isolate 2 and Isolate 4: farm field cv Atora; Isolate 3: susceptible cultivar mix (cvs Avatar, Dariot and Explicit; Isolate 4: susceptible cultivar mix (cvs Avatar, Dariot and Explicit); Isolate 5: clubroot-resistant cultivars Croozer and Crocodile; Isolate 6: clubroot-resistant RNX 193207; Isolate 7: clubroot-resistant cultivar DK Plastic Isolate; 8: susceptible cultivar mix of cvs. Harnas, Avatar, Explicit and Mercedes. Origin of isolates. 1; Hallsberg, 2; Örebro, 3; Simrishamn, 4; Tommelilla, 5–8; Höganäs. ^2^ Regions in Sweden. NA: Närke; SC: Scania. ^3^ Mean value of disease incidence (DI) in clubroot-infested field experimental plots in each county. The incidence of disease in each clubroot-infested field was estimated by randomly choosing 25 plants. The presence of any swelling or gall formation on the root was assumed as proof of *P. brassicae* infection. ^4^ Classification was done according to the differential systems of the European Clubroot Differential set (ECD; [19]) and Somé et al. [20]. A cut-off point of the disease severity index (DSI) of 25% was used to classify plant reactions as resistant or susceptible [20].

#### 3.4.2. Pathotypes of *Plasmodiophora brassicae* in Sweden

The pathotype classification was conducted twice at JKI under controlled greenhouse conditions with comparable results. Four pathotypes were identified as 16/23/28, 16/23/29, 16/31/31, and 16/31/28. The P (+) isolate was found in southern Sweden (Scania) in fields where clubroot-resistant cultivars of winter oilseed rape are grown. The detailed result of the pathotype classification according to the ECD set of Buczacki et al. [19] and Somé et al. [20] is given in Table 4.

### 3.5. Clubroot in Central Europe and Sweden

#### 3.5.1. Disease Occurrence and Severity

During the five-year clubroot study, 81 clubroot-infested fields were identified in the Czech Republic, Germany, Poland, and Sweden. Disease incidence differed within these fields from 3.7% to 100%. For 53.4% of the fields, disease incidence was low (DI ≤30%), whereas moderate clubroot incidences (30% < DI < 60%) were found in 33.3% of the fields, and 17.3% of the fields exhibited a high disease incidence (DI ≥ 60%) (Table 1, Table 2, Table 3 and Table 4). 

#### 3.5.2. Pathotypes of *Plasmodiophora brassicae*

In total, according to Somé et al. [20], six pathotypes of *P. brassicae* were found, with one pathotype in Sweden and four pathotypes in CZ, DE, and PL. Out of the hypothetical eight pathotypes of the system elaborated by Somé et al. [20], P7 and P8 were not found, and P1–P6 were detected, with different frequencies per country (Table 5). The prevailing pathotypes were P1 and P3. In addition, the pathotypes P1 (+), P2 (+), and P3 (+) were also found, with the former one being the most frequent (Table 5). According to Zamani-Noor et al. [27], isolates with such pathotype code overcome the resistance of *B. napus* cv. Mendel.

According to Buczacki et al. [19], virulence analyses differentiated the 84 *P. brassicae*-isolates from Central Europe and Sweden into 42 pathotypes (Table 6). The most common pathotype, 16/31/31, was found in Germany, Poland, and Sweden but not in the Moravian region of the Czech Republic. The second most common pathotype, 16/06/12, was isolated in the Czech Republic (Moravia), Germany, and Poland but was not found in Sweden. Moreover, pathotype 16/15/08 was collected from the Czech Republic and Poland, and pathotype 16/15/31 was present in Germany and Poland (Table 6). Of the 42 virulence patterns based on the ECD differentials, 38 patterns were detected in only one country (Table 6).

The information on field data showed that oilseed rape was grown every 2nd year in 9.2% of the fields and every 3rd year in 51.3% of the fields, and the interval exceeded three years in 39.5% of the fields. The data showed a negative correlation between the frequencies of oilseed rape cultivation in the rotation and the disease incidence. The highest clubroot incidences (DI ≥ 60%) were observed in the fields with a more frequent presence of oilseed rape in the rotation (Figure 5A). Moreover, we observed that soil pH influenced the *P. brassicae* prevalence. Considering all studied fields, a significant negative correlation (r_s_ = -0.2369; *p* = 0.039) was observed between soil pH and disease incidence (Figure 5B).

## 4. Discussion

The study of the occurrence of clubroot in oilseed rape crops clearly showed that in the period from 2016 to 2020, the disease was still present and severe in Moravia (the Czech Republic), Germany, Poland, and Sweden. The disease incidence ranged from 3.7% to 100%, with 51.2% of the fields with DI up to 30%, 32.1% with 30% < DI < 60%, and 16.7% of fields exceeding DI 60%.

Field information from the Czech Republic (Moravia region), Germany, and Poland revealed that in 60.5% of the fields, oilseed rape was grown mainly in rotation once in three years. However, there were fields with oilseed rape grown every two years as well as fields with a time interval of 4–6 years. In one case, oilseed rape was also grown in the same field in the previous year. In Central Europe and Sweden, oilseed rape in a crop rotation is preceded by wheat (*Triticum aestivum* L.) or barley (*Hordeum vulgare* L.), which was also the case in most of the studied fields. The highest clubroot incidences (50–80%) were found in fields where oilseed rape was grown every 2^nd^ year. Furthermore, a significant negative correlation was noticed between clubroot incidence and the frequency of oilseed rape in the rotation. A similar trend was also observed in Germany from 2012 to 2015 [17]. Similarly, Hwang et al. [28] indicated that the rotation sequences, including clubroot-resistant canola, continuous fallow, and the non-host barley crop, decreased gall weight and clubroot severity compared to continuous planting of susceptible canola. In contrast, in Sweden, the rotation of oilseed rape in the studied fields was four and six years. Nevertheless, the incidence of clubroot reached 100% infected plants, even with the rotation of oilseed rape every six years.

Previous studies indicated that clubroot could develop over a wide range of soil pH values, from alkaline to acidic soils. However, the disease incidence and severity varied across different pH values [17,27,28,29,30]. In the current study, we found a significant negative correlation between soil pH and the disease incidence of infested fields across Central Europe. These results are not entirely in line with the previous observations of Řičařová et al. [9], who found a high concentration of *P. brassicae* spores (4.7 × 10^6^ resting spores/g soil) when soil pH was 6.9. To a great extent, this contrasts with previous studies [30,31,32] reporting significant correlations between low soil pH and the occurrence and severity of clubroot. The visible effect of soil alkalinity on the inhibition of root hair infection in cabbage seedlings was reported already in 1945 by Samuel and Garrett [33]. The subsequent design of a simple laboratory method by Crute et al. [34] helped to refine these studies further and showed a decrease in the number of primary plasmodia at pH higher than 7.5 [35]. At pH 5.5, the progression of *P. brassicae* infection was also considerably delayed, but finally, it led to the formation of galls on roots, which was rarely the case at pH 8.0 [35].

Interestingly, our study found a rare pathotype, P5, in Germany at a soil pH close to 7.0. *Plasmodiophora brassicae*, initially severe mainly in acidic soils, might have adapted to a higher soil pH. Its occurrence is nowadays only slightly restricted in soils with neutral pH. It is likely that the pathogen is increasingly adapting to soil pH higher than 7.0. Oilseed rape tolerates a wide range of soil pH values, and 6.5 to 7.0 is considered ideal for most mineral soils, but pH 6.2 is recommended for light-textured soils. Bartomeus et al. [36] found that soil pH was the essential soil factor that explained yield. However, the positive effect of soil pH on seed yield was not detected at high pest levels due to plant damage and the activation of defence mechanisms. The example of the pathotype P5 in soil with neutral pH may also mean that the evolution and adaptation of the new pathotypes may arise from specific soil characteristics. In the study by Řičařová et al. [9], pathotype P4, which was rare, with one isolate among 30 studied in the Czech Republic and Poland, was also detected when soil pH was high (6.8). However, pathotype P5 was detected in soil with a pH of 6.4. Moreover, moderately alkaline soils with pH ranging from 7.1 to 7.6 yielded pathotypes P1 and P3, which were the most common. Therefore, soil pH, as the trigger of the new pathotype evolution, must be treated with caution.

In the present survey, 17 Brassica hosts, including the ECD set [19] and Somé et al. [20], as well as clubroot resistant-oilseed rape cv. Mendel were used to determine the virulence pattern of 84 isolates collected from the Czech Republic, Germany, Poland, and Sweden. We provide clear evidence for a shift towards increased virulence in *P. brassicae* populations from 2016 to 2020 compared with the previous studies. According to the method developed by Somé et al. [20], six pathotypes of *P. brassicae* were found, with one pathotype in SW and four pathotypes in CZ, DE, and PL. This high variation most probably originates from the uneven number of isolates, varying from 8 (Sweden) to 41 (Germany), which caused this difference. Six out of eight hypothetical pathotypes were found, with P1 and P3 as the prevailing ones. The pathotypes overcoming ‘the first-generation clubroot resistance’ available in cv. Mendel were common and constituted a high proportion of the isolates in Poland (60%), Sweden (29%), and Germany (29%). Moreover, 17% of isolates from Germany and 68% of isolates from Poland could overcome the resistance in one, two, or all *B. rapa* species in the ECD differential set.

Quite unexpectedly, the number of P2 isolates in Poland rose dramatically from zero in 2010 to 2012 [9] to 40% in this study. Among these, 16% of the isolates could not overcome the ‘Mendel resistance,’ and 24% overcame this resistance and caused symptoms on *B. napus* cv. Mendel. From 2016 to 2020, pathotypes P4–P6 were still relatively rare in Central Europe. Compared to the previous studies [9,16,17], pathotype P4 increased in the Czech Republic and Poland, whereas single isolates of P5 and P6 were found in Germany and the Czech Republic, respectively, but they were not detected elsewhere. It should be noted that P6 was recorded for the first time in the Czech Republic, which indicates the emergence of a new, rare pathotype in the country.

In Germany, similarly to the previous report [17], differences in disease prevalence and clubroot incidence were observed between and within federal states. Between 2012 and 2015, there were 49 samples from 12 German federal states, but from 2016 to 2020, 41 new clubroot-infested fields were found in seven states. However, regardless of the relative clubroot incidence level, similar pathotypes were identified in both studies, except for the number of highly aggressive isolates that could overcome the resistance of cv. Mendel, which increased between 2016 to 2020.

The classification of the Swedish isolates showed four different pathotypes according to Buczacki et al. [19]. Five of the eight isolates were noted as highly aggressive, indicating that resistance is under pressure. Interestingly, isolate 3 from Scania originated from a farm where a clubroot-resistant cultivar was grown in the field for the second time and was classified as 16/31/31. The clubroot-resistant cultivar Mendel showed erosion of resistance when grown in field soil from this farm in a bioassay [24]. In addition, this isolate was classified as P (+) i.e., highly virulent on cv. Mendel in this study. Moreover, 63% of the isolates were classified as P (+), all of them originating from southern Sweden, whereas the DI of isolates from central Sweden were classified as low (<30).

Clubroot is a severe disease of oilseed rape and other Brassica crops [37,38]. Once established, the resting spores may persist in the soil for up to 17 years [29], and the spread of infected soil with machinery, wind, and water has been demonstrated [15]. Developing and growing clubroot-resistant oilseed rape cultivars is the most effective, economical, and environmentally friendly control strategy for clubroot disease. This study extensively characterized *P. brassicae* populations associated with winter oilseed rape across central Europe and Sweden and examined the variability in the pathogenicity of isolates on 17 Brassica hosts. Collectively, these results reflect the existence of a diverse population of isolates of *P. brassicae* and pathogenicity among the regions. Further studies are needed to monitor new changes in the pathotype population of *P. brassicae.* Such studies are not only academic and serve to increase our knowledge of the current structure and evolution of the pathogen population but also have substantial practical implications. We observed that the use of ‘Mendel resistance’ available in numerous cultivars of oilseed rape in the period of study has led to the selection of pathotypes that can overcome this resistance. To a great extent, the cultivars harbouring the ‘Mendel-resistance genes’ are unsuitable for many fields in Germany, Poland, and Sweden but still important for Moravia in the Czech Republic.

Extensive studies on the effect of the virulence of the pathogen and the soil inoculum density on the disease outbreak [27], as well as the detection of soil-borne inoculum of *P. brassicae* by LAMP [39] or qPCR [24], were performed and constituted important tools for growers choosing a proper cultivar. Frequent cropping of clubroot-resistant winter oilseed rape in infested soil puts varietal resistance under pressure and may eventually lead to resistance breakdown. Recent research provides new knowledge of the ability of *P. brassicae* to evolve new proteins that overcome plant defence. A *P. brassicae*-specific and conserved effector protein PBZF1 that enhances *P. brassicae* virulence was found [40]. The gene coding this effector protein was significantly highly expressed in resting spores. Thus, the breeders and growers should be aware of the rapid development of pathotypes causing high levels of disease in resistant cultivars reported from Canada [16], Germany [17], Sweden [24], and Poland (this study).

## 5. Conclusions

Out of 84 isolates originating from 81 fields in the Czech Republic (Moravia region, 10 fields, 10 isolates), Germany (41 fields, 41 isolates), Poland (25 fields, 25 isolates), and Sweden (5 fields, 8 isolates), as many as 42 pathotypes were designated, using the European Clubroot Differential set [19]. None of these pathotypes was found in all countries. However, two pathotypes were common in three countries, i.e., 16/06/12 in CZ, PL, and DE and 16/31/31 in DE, PL, and SE. Two pathotypes were also common in neighbouring countries, namely, 16/15/08 in CZ and PL and 16/15/31 in DE and PL. According to Somé et al. system [20], there were six pathotypes (P1-P6), with P1 being the most prevalent in Poland and Germany and the only one detected in Sweden. The isolates of *P. brassicae* from Moravia (CZ) belonged mainly to P3. In Poland, the highest number of isolates (60%) overcame the resistance of *B. napus* cv. Mendel, whereas 63% of the isolates from Sweden and 29% of the isolates from Germany overcame cv. Mendel resistance. However, none of the isolates from the Moravia region located in the Czech Republic overcame this resistance. Soil pH and the frequency of oilseed rape in the crop rotation significantly affected the clubroot incidence. Considering all investigated samples, significant negative correlations were found between clubroot incidence and the frequency of oilseed rape in crop rotation, as for clubroot incidence and soil pH.

## Figures and Tables

**Figure 1 pathogens-11-01440-f001:**
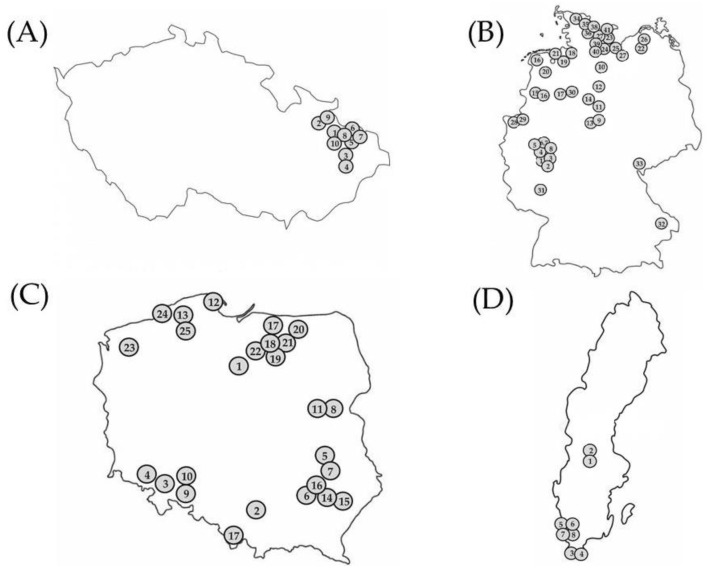
Locations of the origin of 84 *Plasmodiophora brassicae* isolates from four countries (Czech Republic (**A**), Germany (**B**), Poland (**C**), Sweden (**D**)) used for pathotype characterization. Location numbers on the map correspond to the numbers in Tables 1–4. Map sizes do not reflect relative differences between the countries.

**Figure 2 pathogens-11-01440-f002:**
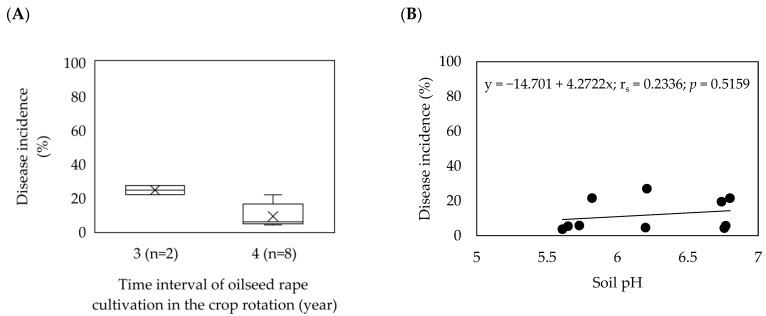
Effect of oilseed rape rotation (**A**) and soil pH (**B**) on the incidence of clubroot (%) in 10 *Plasmodiophora brassicae* infested fields in Moravia, the Czech Republic (2016–2020). (**A**) Box–whisker plots with medians (middle line), mean-value (×), 25% and 75% quartiles, and minimum and maximum values; (**B**) Spearman’s rank correlation coefficient.

**Figure 3 pathogens-11-01440-f003:**
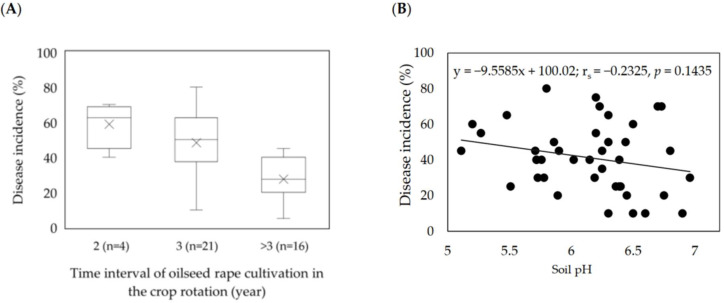
Effect of oilseed rape rotation (**A**) and soil pH (**B**) on the incidence of clubroot (%) in 41 *Plasmodiophora brassicae*-infested fields in Germany (2016–2020)**.** (**A**) Box–whisker plots with medians (middle line), mean-value (×), 25% and 75% quartiles, and minimum and maximum values; (**B**) Spearman’s rank correlation coefficient.

**Figure 4 pathogens-11-01440-f004:**
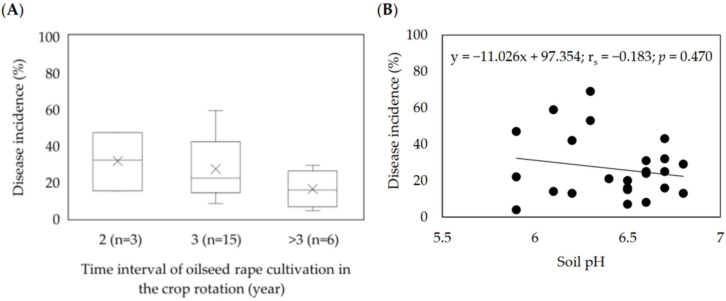
Effect of oilseed rape rotation (**A**) and soil pH (**B**) on the incidence of clubroot (%) in 25 *Plasmodiophora brassicae* infested fields in Poland (2016–2020). (**A**) Box–whisker plots with medians (middle line), mean-value (×), 25% and 75% quartiles, and minimum and maximum values; (**B**) Spearman’s rank correlation coefficient.

**Figure 5 pathogens-11-01440-f005:**
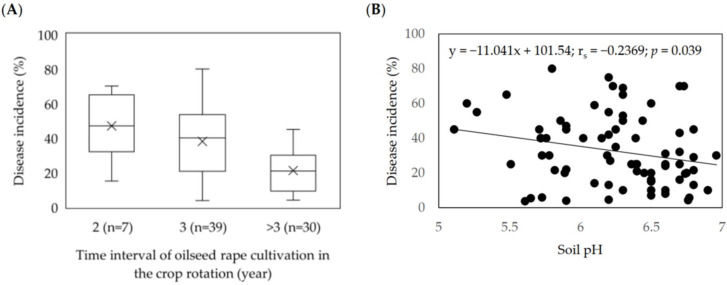
Effect of oilseed rape rotation (**A**) and soil pH (**B**) on the incidence of clubroot (%) in 76 *Plasmodiophora brassicae* infested fields in Central Europe (2016–2020).

**Table 5 pathogens-11-01440-t005:** The frequency distribution of *Plasmodiophora brassicae* pathotypes (%), according to Somé et al. [20], collected in Central Europe (the Czech Republic, Germany, Poland) and Sweden from 2016-2020.

Pathotypes ^1^	Czech Republic (n = 10)	Germany (n = 41)	Poland (n = 25)	Sweden (n = 8)
P1	0	44	8	50
P2	10	2	16	0
P3	60	20	12	0
P4	20	0	4	0
P5	0	5	0	0
P6	10	0	0	0
P7	0	0	0	0
P8	0	0	0	0
P1 (+)	0	24	36	50
P2 (+)	0	0	24	0
P3 (+)	0	5	0	0

^1^ Classification was done according to the differential systems of Somé et al. [20]. A cut-off point of the disease severity index (DSI) of 25% was used to classify plant reactions as resistant or susceptible [20]. (+) represents the *B. napus* cv. Mendel resistance-overcoming pathotypes.

**Table 6 pathogens-11-01440-t006:** The frequency distribution of the *Plasmodiophora brassicae* pathotypes (%), according to Buczacki et al. [19], collected in Central Europe (the Czech Republic, Germany, Poland) and Sweden from 2016-2020.

Pathotypes ^1^	Czech Republic (n = 10)	Germany (n = 41)	Poland (n=25)	Sweden (n = 8)
16/02/30	0	5	0	0
16/03/31	0	2	0	0
16/04/00	10	0	0	0
16/06/25	10	0	0	0
**16/06/12**	**20**	**2**	**4**	**0**
16/06/29	0	0	4	0
16/06/30	0	2	0	0
16/10/00	20	0	0	0
16/12/31	0	2	0	0
16/14/13	10	0	0	0
16/14/15	0	0	8	0
16/14/21	0	2	0	0
16/14/24	10	0	0	0
16/14/30	0	2	0	0
16/14/31	0	2	0	0
**16/15/08**	**10**	**0**	**4**	**0**
16/15/29	10	0	0	0
**16/15/31**	**0**	**2**	**8**	**0**
16/23/28	0	0	0	13
16/23/29	0	0	0	25
16/23/30	0	2	0	0
16/28/30	0	2	0	0
16/30/31	0	2	0	0
16/31/12	0	5	0	0
16/31/28	0	0	0	13
16/31/30	0	20	0	0
**16/31/31**	**0**	**27**	**4**	**50**
17/22/31	0	2	0	0
17/31/31	0	10	0	0
19/31/31	0	2	0	0
20/15/15	0	0	4	0
21/31/15	0	0	4	0
21/31/31	0	0	4	0
24/31/27	0	2	0	0
26/15/31	0	0	4	0
28/15/09	0	0	4	0
30/31/24	0	0	4	0
30/31/26	0	0	4	0
31/15/15	0	0	4	0
31/15/31	0	0	12	0
31/31/15	0	0	8	0
31/31/31	0	0	16	0

^1^ Classification was done according to the differential systems of the European Clubroot Differential set (ECD; [19]). A cut-off point of the disease severity index (DSI) of 25% was used to classify plant reactions as resistant or susceptible [20]. **Bold font** indicates pathotypes common to two or three out of four studied countries.

## Data Availability

Data presented in this study are available upon reasonable request.

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
