# Peer review of "Pathotype Characterization of Plasmodiophora brassicae, the Cause of Clubroot in Central Europe and Sweden (2016–2020)"

_pathogens, 2022, doi:10.3390/pathogens11121440_

Round 1

Reviewer 1 Report

This study analyzes the virulence of the causal pathogen P. brassicae isolated from oilseed clubroot fields occurring in CZ DE, PL, and SW during the 5-year period 2016-2020. The findings are useful for the development of basic research as well as for applications. The paper is worthy of publication in the journal "Pathogens".

 1) Tables 1, 2, 3, and 4, "Mean value of DI in clubroot-infested field (%)": Why are the DIs for each strain not shown in the Tables? Also, many "mean values of DIs" are shown in the Tables, but it is unclear from what each was calculated. Either the DI for each isolate should be shown in each Table, or the definition of "Mean value of DI in clubroot-infested field (%)" should be given in the footnote.

2) Tables 2, 3, and 4, "pathotype classification": Please show the footnotes in Table 5, as well as in Tables 2, 3, and 4.

Reviewer 2 Report

The present manuscript (ID: pathogens-2050005) reports the pathotype characterization of 84 Plasmodiophora brassicae isolates collected from Czech Republic, Germany, Poland, and Sweden. Some interesting new information has been provided on the better understanding for clubroot pathogens in oilseed rape in Europe. It is also a useful funding that rotation and soil pH indicated correlations with clubroot incidence. Since the manuscript has been well prepared and fits the journal’s scope, the publication of the manuscript might be considered if it would be properly revised, including suggestions as:

Please provide more information on the rotation regime, i.e. the previous crops.

How the soil pH determined for the DI investigation sites?

Add the statistical analyses method for correlation.

In table 2-4, the column “Mean value of DI in clubroot-infested field” is for isolate/site? or other else?

The statistical evidences in Figure 2, 3, and 4, except in Figure 5, were not sufficient to support the conclusion in line “26-28” , which explanation is needed.
